# Metabolic Regulation of Endothelial Cells: A New Era for Treating Wet Age-Related Macular Degeneration

**DOI:** 10.3390/ijms25115926

**Published:** 2024-05-29

**Authors:** Xirui Chen, Yang Xu, Yahan Ju, Ping Gu

**Affiliations:** 1Department of Ophthalmology, Ninth People’s Hospital, Shanghai Jiao Tong University School of Medicine, Shanghai 200011, China; sirichen1999@sjtu.edu.cn (X.C.);; 2Shanghai Key Laboratory of Orbital Diseases and Ocular Oncology, Shanghai 200011, China

**Keywords:** wet age-related macular degeneration, endothelial cell metabolism, therapeutic strategy, angiogenesis

## Abstract

Wet age-related macular degeneration (wet AMD) is a primary contributor to visual impairment and severe vision loss globally, but the prevailing treatments are often unsatisfactory. The development of conventional treatment strategies has largely been based on the understanding that the angiogenic switch of endothelial cells (ECs) is mainly dictated by angiogenic growth factors. Even though treatments targeting vascular endothelial growth factor (VEGF), like ranibizumab, are widely administered, more than half of patients still exhibit inadequate or null responses, suggesting the involvement of other pathogenic mechanisms. With advances in research in recent years, it has become well recognized that EC metabolic regulation plays an active rather than merely passive responsive role in angiogenesis. Disturbances of these metabolic pathways may lead to excessive neovascularization in angiogenic diseases such as wet AMD, therefore targeted modulation of EC metabolism represents a promising therapeutic strategy for wet AMD. In this review, we comprehensively discuss the potential applications of EC metabolic regulation in wet AMD treatment from multiple perspectives, including the involvement of ECs in wet AMD pathogenesis, the major endothelial metabolic pathways, and novel therapeutic approaches targeting metabolism for wet AMD.

## 1. Introduction

Age-related macular degeneration (AMD), a degenerative condition of the retina, has emerged as the predominant cause of visual impairment and profound vision decline, particularly in the aged demographic [1,2]. Forecasts indicate a surge in AMD prevalence, with patient numbers anticipated to escalate from nearly 200 million in 2020 to around 300 million by 2040 [3]. AMD manifests in two primary forms, dry and wet, with the latter, though less prevalent, being chiefly responsible for the significant instances of visual deterioration. The pathogenesis of wet AMD is intricate, currently understood to be the result of genetic, age-related, and environmental interplay. At the core of its pathophysiology lies the formation of choroidal neovascularization (CNV), where abnormal blood vessels breach Bruch’s membrane, growing towards the retina, accompanied by exudation, hemorrhage, and a series of pathological changes, leading to gradual central vision decline and significantly impacting the quality of life for affected individuals [4,5,6].

The prevailing therapeutic approach to wet AMD in clinical settings primarily aims to mitigate pathological neovascularization, principally through intravitreal administration of anti-vascular endothelial growth factor (VEGF) biologics such as ranibizumab [7], bevacizumab [8], and aflibercept [9], which have demonstrated substantial efficacy. However, numerous challenges and limitations persist, as over half of patients exhibit inadequate or null responses to these treatments [10,11], highlighting the need for alternative therapeutic strategies to effectively manage CNV.

In recent years, anti-angiogenic therapies centered around growth factors have been critically re-evaluated, with researchers increasingly recognizing the significant role of endothelial cell (EC) metabolic regulation in angiogenic diseases [12,13,14,15]. Under normal conditions, ECs remain quiescent, maintaining metabolic equilibrium to preserve homeostasis. However, in pathological conditions such as wet AMD, ECs transition into a state of excessive angiogenesis, undergoing metabolic reprogramming to meet their increased demands for energy and biomass [16]. Recent studies have highlighted the role of EC glycolysis and its key regulatory enzyme, 6-phosphofructo-2-kinase/fructose-2,6-biphosphatase 3 (PFKFB3), in the regulation of angiogenesis [17,18,19]. Imbalances in other endothelial metabolic pathways such as lipid metabolism and glutaminolysis have also been shown to impact neovascularization [20,21]. Therefore, restoring homeostasis by targeted modulation of EC metabolism holds promise as a novel strategy for treating wet AMD and other angiogenic diseases. In this review, we systematically discuss the potential applications of EC metabolic regulation in wet AMD treatment from multiple aspects, including the involvement of ECs in wet AMD pathogenesis, the major endothelial metabolic pathways, and novel therapeutic approaches targeting metabolism for wet AMD, aiming to achieve new breakthroughs in wet AMD treatment.

## 2. The Involvement of ECs in Wet AMD Pathogenesis

### 2.1. Relationship between EC Dysfunction and Wet AMD

As the major cellular components of blood vessel walls, ECs play critical roles in maintaining vascular homeostasis and regulating angiogenesis. Under normal conditions, tight junctions between choroidal microvascular ECs form the blood–retinal barrier, maintaining homeostasis in outer retinal tissues. However, early loss of choroidal microvascular ECs has been observed in wet AMD patients [22], disrupting the integrity of the blood–retinal barrier and suggesting that EC dysfunction may be one of the initiating factors for wet AMD development.

Studies have shown significantly increased circulating ECs in wet AMD patients, which exhibit pronounced functional abnormalities [23]. One manifestation of EC dysfunction is impaired endothelium-dependent vasodilation, demonstrated by decreased nitric oxide (NO) and increased endothelin-1 (ET-1) in the eyes and circulation of patients [24,25]. Furthermore, EC dysfunction is also reflected in enhanced proinflammatory responses, with increased expression of inflammatory factors such as interleukin-1β (IL-1β), tumor necrosis factor-α (TNF-α), and monocyte chemoattractant protein-1 (MCP-1) in CNV in wet AMD patients [26,27]. Elevated reactive oxygen species (ROS) levels, which are closely associated with EC dysfunction, can damage ECs through mechanisms such as inducing EC apoptosis, accelerating EC senescence, and activating inflammatory signaling pathways [28,29,30,31]. Collectively, EC dysfunction plays multifaceted critical roles in the pathogenesis of wet AMD, from disrupting the integrity of the blood–retinal barrier to promoting inflammatory responses and oxidative stress changes that together drive disease progression. As shown in the figure below, the retinal and choroidal structures of wet AMD patients exhibit pathological alterations, particularly the abnormalities in the choriocapillaris endothelial cells (Figure 1). 

### 2.2. Evidence for the Involvement of EC Metabolic Dysfunction in the Pathogenic Mechanisms of Wet AMD

Mounting evidence suggests that EC metabolic dysfunction may play a key role in the pathogenic mechanisms of wet AMD. While direct evidence linking the two is still limited, several indirect lines of evidence strongly support a close association between EC metabolic abnormalities and pathological processes in wet AMD. Two independent single-cell transcriptomic analyses provided critical evidence in this regard. The first study found significant changes in the expression of multiple metabolism-related genes (such as glycolysis, oxidative phosphorylation, and nucleotide biosynthesis) in choroidal vascular ECs from wet AMD patients compared to normal controls [32]. The second study further revealed increased amino acid metabolism and angiogenesis sensitivity as well as stress responses to amino acid deprivation, including activation of the General Control Nonderepressible 2 (GCN2) signaling pathway and changes in amino acid and derivative metabolism in ECs from wet AMD patients [33]. These findings indicate widespread metabolic abnormalities in ECs from wet AMD patients, which could represent core components of the pathogenic mechanisms. This not only deepens our understanding of wet AMD pathogenesis but also lays a solid foundation for further exploring the specific roles of EC metabolic dysfunction in wet AMD.

Some animal models of wet AMD also involve changes in EC metabolism. In a laser-induced CNV mouse model, it was found that Yes-Associated Protein (YAP) upregulates PFKFB3 expression to drive a significant increase in EC glycolytic activity, while inhibition of YAP or PFKFB3-mediated glycolysis can markedly alleviate CNV formation [19]. This suggests that enhanced glycolytic metabolism in ECs may be an important factor promoting pathological angiogenesis in wet AMD. Another study using the same model further revealed that increased glycolysis leads to significantly elevated lactate levels, which can stimulate macrophage VEGF release to further promote endothelial tube formation and CNV development [34]. These findings indicate that abnormal EC glycolytic metabolism may play important roles in the pathogenic processes of wet AMD through direct (promoting EC proliferation and tubulogenesis) and indirect (lactate paracrine effects on macrophages) mechanisms.

While metabolomic studies of ECs from wet AMD patients are still limited, analyses of blood or ocular metabolites have provided valuable clues. For example, one study found significant differences in 53 plasma lipids (including glycerophospholipids, sphingolipids, glycerides, and fatty acids) between wet AMD patients and controls [35]. Another study reported significant differences in 48 plasma metabolites (mainly lipid metabolism, especially glycerophospholipid metabolism) between different disease severity stages of late AMD [36]. These findings suggest an important role for lipid metabolic dysfunction in wet AMD pathogenesis. In addition, analysis of aqueous humor metabolites revealed glucose metabolism-related changes in wet AMD patients, including increased tricarboxylic acid cycle metabolite substrates and decreased α-ketoglutarate levels [37]. While these studies did not directly analyze EC metabolism, the critical roles of ECs in wet AMD pathology and the close relationship between EC metabolism and vascular abnormalities lead us to hypothesize that some of these metabolic alterations may be associated with EC metabolic dysfunction, providing clues for further investigating the role of EC metabolism in wet AMD.

Metabolic dysfunction has also played important roles in the pathogenesis of various retinal vascular diseases, providing some insights into potential mechanisms in wet AMD. For example, hyperglycemia-induced EC glycolysis enhancement and mitochondrial dysfunction in diabetic retinopathy models are considered important causes of EC dysfunction and pathological angiogenesis [38]. Another study reported changes in glutamine metabolism in an oxygen-induced retinopathy model of preterm infants, emphasizing its importance in developmental and pathological angiogenesis [21]. Considering the similarities in vascular abnormalities between wet AMD and these diseases, this suggests metabolic dysfunction could represent a common pathogenic basis across various retinal vascular diseases including wet AMD.

In conclusion, while direct evidence linking EC metabolic dysfunction to wet AMD is still limited, the indirect evidence, insights from related diseases, and the importance of metabolic dysfunction in EC function lead us to hypothesize that EC metabolic abnormalities may represent an important component of wet AMD pathogenesis. Future studies providing more direct evidence of the relationship between EC metabolic states and wet AMD are needed to further elucidate its roles and mechanisms in disease occurrence and progression, providing a theoretical basis for developing new intervention strategies.

## 3. Major Metabolic Pathways and Their Regulation in ECs

ECs lining the inner surfaces of blood vessels play pivotal roles in maintaining vascular homeostasis and regulating angiogenesis. Metabolic activity is critically important for EC function, as it provides the necessary energy and biosynthetic precursors to support various cellular processes. The principal metabolic pathways in ECs include glycolysis, mitochondrial metabolism, lipid metabolism, and amino acid metabolism (Figure 2). These metabolic routes are highly interconnected and collectively regulate both physiological and pathological processes in ECs. Understanding the regulation and interplay of these major metabolic pathways in ECs is crucial, as disturbances in EC metabolism have emerged as an important contributor to the pathogenesis of various vascular diseases, including wet AMD. The following sections will delve deeper into the specific roles and regulation of these key metabolic routes in ECs.

### 3.1. Glycolysis and Its Branches

Despite their direct exposure to an oxygenated blood environment, ECs obtain ATP mainly through glycolysis rather than mitochondrial oxidative phosphorylation [39]. Studies have shown that the glycolytic rate in ECs is extremely high, over 200 times that of glucose oxidation and fatty acid oxidation, similar to many cancer cells [17,40]. The preference for aerobic glycolysis in ECs may be attributed to two factors. Firstly, glycolysis can rapidly and abundantly generate ATP to meet the energetic demands of EC sprouting and adaptation to hypoxic environments [16,41,42]. Secondly, glycolysis provides important metabolites for biosynthetic pathways such as nucleotide synthesis, protein glycosylation, and amino acid synthesis, thereby promoting various physiological functions of ECs [16,43].

ECs uptake glucose via glucose transporters (GLUTs) to initiate glycolysis [44], which is then catalyzed through a series of enzymes to break down glucose into pyruvate and ultimately produce ATP and lactate. Glycolysis in ECs is tightly regulated. Hypoxia-inducible factor-1α (HIF-1α) is a key regulator of glycolysis. Under hypoxic conditions, stabilized HIF-1α translocates into the nucleus, where it forms a heterodimer with HIF-1β and binds to hypoxia response elements (HREs) in the promoter regions of downstream target genes such as GLUT1, hexokinase (HK), and phosphofructokinase (PFK), upregulating these key glycolytic enzymes and accelerating glycolysis [45,46,47]. Additionally, PFKFB3 catalyzes the generation of fructose-2,6-bisphosphate (F2,6P_2_) from fructose-6-phosphate (F6P), an allosteric activator of PFK, thereby functioning as a rate-limiting enzyme of glycolysis. Pharmacological inhibition or genetic silencing of PFKFB3 impairs EC sprouting in vitro and retinal angiogenesis in vivo [17,18]. In contrast, overexpression of PFKFB3 increases glycolysis and induces an endothelial tip cell phenotype, even suppressing the Notch Receptor 1 (NOTCH1) signaling in retinal vascular development [17]. Recent studies further revealed the proangiogenic roles of YAP-driven glycolysis and its metabolic end product lactate in CNV [19,34]. In summary, EC glycolysis is finely tuned and plays crucial roles in both vascular formation and homeostasis.

The pentose phosphate pathway (PPP) is an ancillary branch of the glycolysis pathway that utilizes glucose-6-phosphate (G6P) from glycolysis to generate reduced nicotinamide adenine dinucleotide phosphate (NADPH) and ribose-5-phosphate (R5P) [48]. The PPP consists of an irreversible oxidative branch and a reversible nonoxidative branch catalyzed by glucose-6-phosphate dehydrogenase (G6PD) and transketolase, respectively [49]. In the oxidative branch, G6PD catalyzes the conversion of G6P to NADPH, which serves as a cofactor for antioxidant systems such as glutathione reductase to clear ROS in ECs [50]. Additionally, NADPH participates in fatty acid and NO synthesis to promote EC proliferation, migration, and angiogenesis [51]. The nonoxidative branch produces the key intermediate R5P for nucleotide biosynthesis [40]. Inhibition of any rate-limiting enzyme in the PPP decreases EC viability [49], while G6PD overexpression stimulates EC proliferation, migration, and tube formation [52].

The hexosamine biosynthesis pathway (HBP) is another important branch of glycolysis. Though accounting for a small proportion of glucose catabolism, it plays critical roles in protein post-translational modification processes [53,54]. The HBP utilizes glycolytic intermediates to synthesize N-acetylglucosamine for O- and N-glycosylation of proteins [54]. Interestingly, the functions of multiple key proteins involved in angiogenesis, such as Notch and VEGFR2, depend on their glycosylation status in ECs [55,56,57,58].

### 3.2. Mitochondrial Metabolism

Compared to other cell types, ECs contain fewer mitochondria and exhibit relatively lower oxygen consumption [59]. Under physiological glucose levels, only a small amount of pyruvate enters the mitochondria to participate in the tricarboxylic acid (TCA) cycle, generating reduced nicotinamide adenine dinucleotide (NADH) and reduced flavin adenine dinucleotide (FADH2) to drive the electron transport chain and ATP synthesis [17]. Despite this, ECs maintain their oxidative metabolic capacity under conditions of glycolytic inhibition or stress [60]. Importantly, mitochondria play significant signaling and regulatory roles in ECs beyond ATP generation. Mitochondria-derived ROS act as important signaling molecules involved in endothelial function regulation [61,62,63]. Moderate levels of ROS are crucial for maintaining endothelial homeostasis, while excessive ROS can lead to endothelial dysfunction. Additionally, mitochondria influence multiple aspects of endothelial function, including vascular permeability, cell migration, proliferation, and angiogenesis through regulation of calcium ion homeostasis [64].

### 3.3. Lipid Metabolism

Fatty acids (FAs) taken up by ECs from the blood circulation or synthesized de novo by fatty acid synthase (FASN) undergo fatty acid oxidation (FAO) for utilization. While FAO only accounts for less than 5% of total ATP production in ECs, sprouting tip cells depend on FAO during angiogenic vascular formation [43,65]. Studies have shown that EC FA uptake is mainly mediated by fatty acid binding protein 4 (FABP4), and VEGF can upregulate FABP4 expression to promote EC proliferation [66,67]. Carnitine palmitoyltransferase 1 (CPT1) is required for fatty acid entry into mitochondria, functioning as the rate-limiting enzyme that adds an acetyl-CoA group to FAs [68]. Within mitochondria, FAs undergo β-oxidation to generate acetyl-CoA, which is then fed into the TCA cycle. FA-derived acetyl-CoA along with substrate replenishment sustains the TCA cycle to produce aspartate and glutamate, the precursors for deoxynucleotide triphosphate (dNTP) synthesis essential for DNA replication in proliferating tip cells [43]. Specific deletion of CPT1 in ECs reduces EC proliferation and induces defects in vascular sprouting in vitro and in vivo [43]. Additionally, ECs participate in cholesterol metabolism regulation. ECs express cholesterol efflux transporters ATP-binding cassette transporter A1 (ABCA1) and ATP-binding cassette transporter G1 (ABCG1) to transport excess intracellular cholesterol to apolipoprotein A-I (apoA-I) and high-density lipoprotein (HDL), exerting anti-atherosclerotic effects [69,70].

### 3.4. Amino Acid Metabolism

Amino acid metabolism also plays a crucial role in maintaining vascular function and development in ECs. Glutamine is the most consumed amino acid in ECs, which is degraded into glutamate and ammonia by glutaminase (GLS) [21]. This process not only provides substrate for the TCA cycle to support EC growth and vascular dilation, but also participates in glutathione production, maintenance of intracellular redox homeostasis, and promotion of angiogenesis [21,71]. When glutamine is deficient, aspartate alone can partially rescue EC defects, and silencing the aspartate synthesis enzyme impairs EC proliferation, indicating the importance of aspartate in compensating for glutamine metabolism [21]. Notably, in addition to known glutamine catabolism, glutaminase displays unknown activity through Ras homolog family member J (RHOJ) palmitoylation during pathological angiogenesis involving EC migration [72]. Arginine is an important source for endothelial nitric oxide synthase (eNOS) to generate the vasoprotective NO in ECs, whose depletion leads to eNOS dysfunction, reduced NO production by ECs, and impaired vascular relaxation function [73,74,75]. In addition, ECs can take up serine or convert the glycolytic intermediate 3-phosphoglycerate into serine. Serine can further be converted to glycine in ECs, participating in nucleotide biosynthesis and redox homeostasis regulation [76].

## 4. Metabolic Regulation of ECs and Treatment of Wet AMD

Given the important role of EC metabolic dysfunction in the pathogenesis of wet AMD, strategies targeting EC metabolic pathways hold promise as novel approaches to wet AMD treatment. By modulating key metabolic pathways in ECs, such as glycolysis and lipid metabolism, we may restore normal EC function, inhibit CNV formation, and ultimately achieve wet AMD treatment. In the following sections, we will discuss therapeutic strategies targeting different metabolic routes and their prospects in wet AMD treatment in detail (Table 1).

### 4.1. Targeting Glycolysis for Wet AMD Treatment

Given the critical role of glycolysis and its central regulator, PFKFB3, in angiogenesis management, targeting this pathway and protein offers a promising strategy for anti-angiogenic therapy in wet AMD.

3-(3-pyridinyl)-1-(4-pyridinyl)-2-propen-1-one (3PO) is a small molecule inhibitor that exerts its effects by selectively inhibiting the activity of PFKFB3. Studies have shown that 3PO can inhibit pathological angiogenesis in a CNV model and oxygen-induced retinal degeneration [18]. Notably, 3PO treatment reduced glycolytic flux by approximately 40%, weakening EC proliferation and migration capabilities. However, this process did not cause EC death but rather promoted a reversible quiescent state, indicating that PFKFB3 blockade decreases the high metabolism triggered during the transition of ECs from quiescence to proliferation and migration without affecting their basal requirements [18]. Additionally, combining 3PO with anti-VEGFR2 monoclonal antibody DC101 was found to enhance inhibition of CNV in mice [18]. 3PO can reduce skin inflammation and inflammation-associated vascular density, demonstrating its inhibitory effects in inflammatory processes [18,97], independent of its inhibition of glycolysis-derived effects.

PFK15 is another selective PFKFB3 inhibitor that displays higher activity and selectivity compared to 3PO [98]. Studies showed that PFK15 can inhibit glucose metabolism in hemangioma-derived endothelial cells (HemECs), thereby impacting their angiogenic and migratory capabilities [77]. Furthermore, PFK15 also exhibited potential for promoting vascular structural and functional normalization [99]. Based on PFK15, researchers obtained a molecule with higher specificity and efficacy, PFK158, as a PFKFB3 small molecule inhibitor [78,79,100], which was the first PFKFB3 inhibitor evaluated in a Phase I clinical trial (NCT02044861), demonstrating its immense potential for anti-angiogenic therapy.

2-deoxy-D-glucose (2-DG) is one of the most extensively studied hexokinase (HK) inhibitors. It is a glucose analog where the hydroxyl group on the second carbon is replaced by hydrogen [101]. Due to this structural similarity, 2-DG can be recognized and taken up by glucose transporters and phosphorylated by hexokinase, but its phosphorylated product, 2-DG-6-phosphate (2-DG-6-P), cannot be further metabolized and accumulates in cells, thereby interfering with the glycolytic pathway [102]. Studies showed that 2-DG can suppress filopodia formation and induce disruption of F-actin filaments [103], thereby inhibiting angiogenesis in vitro and in vivo [18,80,81]. In cancer treatment, 2-DG is often combined with conventional therapies such as radiotherapy or chemotherapy to potentially enhance therapeutic effects [101,104]. While some scholars argue that 2-DG’s inhibition of angiogenesis may be through suppression of N-glycosylation rather than glycolysis, the exact mechanism remains unclear. Moreover, 2-DG may also affect normal cell energy metabolism and cause damage to normal tissues, limiting its development due to nonspecific effects. Future research needs to better understand the mechanisms of 2-DG to optimize its efficacy and safety.

Shikonin is a naphthoquinone compound extracted from the roots of Lithospermum erythrorhizon. Recent research has found that shikonin can act as an inhibitor of pyruvate kinase M2 (PKM2) and exert anti-angiogenic effects through regulating EC glycolysis [82,105]. PKM2 is an isoform of pyruvate kinase (PK), another key rate-limiting enzyme in the glycolytic pathway that converts phosphoenolpyruvate (PEP) to pyruvate with concomitant ATP generation [106]. It plays an important role in angiogenesis in ECs, and studies have shown that PKM2 regulates angiogenesis by modulating intercellular connections and collective migration between ECs [82]. Gene silencing or pharmacological inhibition of PKM2 can reduce the ATP required for connections between ECs, affecting the endocytosis and transport of VE-cadherin at cell junctions, leading to instability of intercellular connections and weakened angiogenesis [82]. Furthermore, shikonin not only inhibits PKM2 but can also impact the expression of angiogenic factors and related downstream signaling pathways such as VEGF and PI3K/AKT, conferring an advantage of multi-target synergistic effects [107,108,109].

In addition to the key rate-limiting glycolytic enzymes mentioned above, such as PFKFB3, HK2, and PKM2, other glycolytic rate-limiting enzymes like phosphoglucose isomerase (PGI) and lactate dehydrogenase (LDH) may also play certain roles in the development of angiogenesis-related diseases, including wet AMD, by regulating EC glycolysis. This is worthy of more attention. However, there are still some limitations to targeting glycolysis for treating wet AMD. On the one hand, while the roles of various glycolytic rate-limiting enzymes have been preliminarily verified in some angiogenesis-related disease models, their specific mechanisms in wet AMD remain to be further elucidated. On the other hand, highly specific, efficient, and low-toxic glycolytic rate-limiting enzyme inhibitors are still lacking, and existing inhibitors may have off-target effects or impact normal cell metabolism. In addition, excessive inhibition could lead to serious side effects as glycolysis is closely related to many physiological processes in the body. Therefore, future research needs to develop specific, efficient, and safe glycolytic rate-limiting enzyme inhibitors based on in-depth understanding of EC glycolytic metabolism regulation, and conduct rigorous preclinical and clinical studies to ultimately achieve the goal of targeting glycolysis for wet AMD treatment.

### 4.2. Targeting Mitochondrial Function for Wet AMD Treatment

While mitochondria serve as an important site of energy production in cells, their role in regulating angiogenesis has received little attention due to the low correlation between ATP and blood vessel growth. With deeper research, it is increasingly recognized that mitochondria perform vital signaling and regulatory functions in cells beyond ATP generation, such as ROS production and calcium ion homeostasis, providing novel insights into modulating their effects on angiogenesis from other mitochondrial aspects.

Indeed, many compounds that interfere with mitochondrial function, such as respiratory chain inhibitors or ROS scavengers, have demonstrated promising anti-angiogenic effects in experimental studies. Ubiquinol-cytochrome c reductase binding protein (UQCRB) is a subunit of mitochondrial complex III, and research shows that UQCRB enhances VEGFR2 signaling and promotes VEGF-dependent vascular growth by increasing mitochondrial ROS levels [84]. Terpestacin is a naturally occurring bisorbicillinoid molecule found to exert general inhibitory effects on vascular growth in early research [110]. Further studies confirmed that terpestacin can specifically bind UQCRB to effectively suppress VEGF-induced EC proliferation, migration, and tube formation, fundamentally blocking vascular growth [83,84,85]. Additionally, terpestacin in combination with bevacizumab showed strong synergistic effects against tumor angiogenesis [84].

MitoQ is an antioxidant that targets mitochondria and may have therapeutic potential for angiogenesis-related diseases [86]. MitoQ was designed by covalently linking a more potent antioxidant, coenzyme Q10 (Q10), to a triphenylphosphonium cation to facilitate its uptake into mitochondria across cellular membranes [111]. Once inside mitochondria, MitoQ can clear ROS and protect mitochondria from oxidative stress damage. MitoQ induces HIF1α degradation by inhibiting tumor necrosis factor receptor-associated protein 1 (TRAP1) activity, downregulating its regulated pro-angiogenic factors; experiments showed MitoQ reduced avascular zones and new blood vessels in an ischemic retinal animal model [86]. Studies also demonstrated that MitoQ restores endothelial barrier integrity by preventing vascular endothelial cadherin (VE-cadherin) degradation and cytoskeletal remodeling of actin filaments, as well as reducing inflammatory responses of nuclear factor kappa-light-chain-enhancer of activated B cells (NF-κB) and NOD-like receptor protein 3 (NLRP3) inflammasomes in ECs [112]. Moreover, MitoQ maintains mitochondrial function by decreasing ROS production and excessive autophagy [112].

As shown above, some research has indicated that targeting mitochondria may be a promising anti-angiogenic strategy, but the precise mechanisms of mitochondria in angiogenesis remain unclear, and directly manipulating mitochondrial function for therapeutic purposes faces many challenges. For example, the effects of mitochondrial ROS at different levels on ECs are complex, and current drugs’ ability to regulate its levels is not precise enough [113]; mitochondrial biological regulators may act differently under varying conditions [63]; and the relationship between mitochondrial calcium signaling coupling and angiogenesis regulation requires further study [64]. Furthermore, the safety and efficacy of existing mitochondrial-targeting drugs in clinical application need further optimization. Overall, while targeting mitochondria is viewed as a potential anti-angiogenic approach, limitations in mechanisms and drug development require deeper investigation to overcome barriers and improve the feasibility and efficiency of practical application.

### 4.3. Targeting Lipid Metabolism for Wet AMD Treatment

A new direction in the development of anti-angiogenic therapies focuses on targeted interventions at key nodes of fatty acid metabolism. For example, important enzymes or pathways involved in fatty acid synthesis and oxidation, through precise regulation of these targets’ activities, may achieve better control over pathological vascular growth and provide new feasible options for relevant diseases such as wet AMD.

Orlistat is a selective FASN inhibitor primarily used to treat obesity. In recent years, research has found that orlistat exerts multi-pathway inhibition of angiogenesis and has shown good effects in improving pathological neovascularization of the eye and anti-tumor angiogenesis [87,88,114,115]. Specifically, orlistat can promote the expression of anti-angiogenic VEGF isoforms such as VEGF165b while downregulating levels of pro-angiogenic factors [88,115]. Additionally, orlistat can upregulate mechanistic target of rapamycin (mTOR) acetylation and reduce the pro-angiogenic activity of mTOR complex 1 (mTORC1), thereby inhibiting EC proliferation [87]. On the other hand, FASN silencing can regulate the expression and activity of matrix metalloproteinase-9 (MMP-9), affecting the bioavailability of VEGF in the extracellular environment and suppressing tumor angiogenesis [116].

Etomoxir is an irreversible inhibitor of CPT1 that can inhibit mitochondrial long-chain FAO [68]. In vitro experiments have shown that etomoxir treatment can reduce FAO levels and proliferative capacity in ECs without affecting cell migration [43]. In a mouse model of retinal vascular development, etomoxir-treated mice exhibited retinal vascular developmental defects similar to CPT1a knockout EC mice, such as reduced branch points [43]. Moreover, in an early mouse model of retinal disease, etomoxir exerted anti-angiogenic effects by reducing pathological vascular tumor formation [43].

We also note that various 3-hydroxy-3-methylglutaryl-coenzyme A (HMG-CoA) reductase inhibitors (statins) exhibit anti-angiogenic properties, among which treatment doses of pitavastatin have been proven to improve CNV in rats and mice [89,90,117,118]. However, whether statins have a protective effect in wet AMD remains controversial in clinical studies [119]. More clinical research is still needed to evaluate the precise protective effects and optimal dosing strategies of statins in wet AMD. Meanwhile, the anti-angiogenic mechanisms of statins also require further exploration into whether they relate to regulating EC lipid metabolism. Overall, targeting key nodes in the regulation of EC lipid metabolism may provide new therapeutic directions for wet AMD. But further optimization of targeted drugs and dosing regimens is still needed before clinical application to achieve better therapeutic effects.

### 4.4. Targeting Other Metabolic Pathways for Wet AMD Treatment

#### 4.4.1. Pentose Phosphate Pathway

6-Aminonicotinamide (6-AN) is a classic non-competitive inhibitor of the PPP [92]. It is a nicotinamide derivative that acts by interfering with the oxidative phase of the PPP. Specifically, 6-AN can inhibit G6PD, a key enzyme in the oxidative phase of the PPP. G6PD is the main source of cellular NADPH, which is a cofactor required for eNOS to generate NO. A decrease in G6PD activity means reduced levels of available NO in ECs, affecting EC function [91]. Furthermore, tyrosine phosphorylation of G6PD correlates with Akt phosphorylation mediated by VEGF and EC migration [120]. Increased G6PD activity promotes EC proliferation, migration, and tubular structure formation [52,91]. While 6-AN has shown certain anti-angiogenic effects in laboratory research, its effects are non-specific and may affect other cellular metabolic pathways, leading to unpredictable impacts. Further research and clinical validation are needed to evaluate its potential for clinical application.

#### 4.4.2. Glycosylation

VEGFR2 is an important EC growth signaling protein that plays a crucial role in angiogenesis by promoting EC proliferation, migration, and permeability functions [121,122]. Studies have found that VEGFR2 is a highly N-glycosylated receptor protein, and its N-glycosylation status can regulate VEGFR2 ligand binding and downstream signal transduction [57]. Itraconazole is a widely used antifungal drug and recent research has discovered that it has powerful anti-angiogenic activity [93,123,124]. Specifically, itraconazole can cause accumulation of immature N-glycans on VEGFR2, thereby inhibiting VEGFR2 transport and signal transduction and exerting anti-angiogenic biological effects [123]. In laboratory studies, itraconazole demonstrated anti-angiogenic effects in ECs derived from multiple vascular sources [93,94], and its administration in the vitreous cavity effectively suppressed the development of laser-induced CNV in rats [95]. As a glycosylation regulator, the application prospects of itraconazole in wet AMD and other related diseases deserve in-depth research.

#### 4.4.3. Glutaminolysis

GLS plays an important role in regulating EC function and promoting angiogenesis [21,72,96]. Currently, several GLS inhibitors have shown potential anti-angiogenic effects. Among them, 6-Diazo-5-oxo-L-norleucine (DON) is a non-selective GLS inhibitor, while Bis-2-(5-phenylacetamido-1,3,4-thiadiazol-2-yl) ethyl sulfide (BPTES) and CB-839 are more selective GLS1 inhibitors [96]. Studies have found that GLS1 can promote human EC growth and survival through multiple pathways, such as enhancing cell proliferation, migratory ability, and maintaining redox balance, which have been observed in venous, arterial, and microvascular ECs [96]. These GLS/GLS1 inhibitors may inhibit angiogenesis by blocking GLS1 regulation of EC function [96]. Moreover, CB-839 can also weaken postnatal and oxygen-induced vascular growth in a mouse model of retinal pathology [21,125]. Taken together, GLS inhibitors have performed well in preclinical research, but the physiological functions of GLS in normal tissues also require further clarification to avoid unnecessary toxic side effects.

### 4.5. Potential Advantages of EC Metabolic Regulation Strategies in Wet AMD Treatment

EC metabolic regulation strategies may have the following advantages compared to conventional anti-VEGF therapies for the treatment of wet AMD:

Fundamental mechanism: In contrast to anti-VEGF therapy, which primarily blocks a single VEGF signaling pathway to inhibit angiogenesis, metabolic regulation strategies in ECs can target key metabolic pathways such as glycolysis, mitochondrial respiration, and lipid metabolism. This multi-target regulation approach can more comprehensively influence the physiological functions of ECs, fundamentally ameliorating their abnormal metabolic state, and thereby more effectively inhibiting the formation and progression of CNV.

Higher safety: In recent years, the development of ophthalmic biomedical materials has advanced rapidly [126]. Through the utilization of advanced biomaterials and delivery technologies, such as nanocarriers and hydrogels, the targeted delivery of drugs that modulate EC metabolism can be achieved. By integrating the EC metabolic regulation strategies of these drug delivery systems, it is possible to more precisely target the key cell types involved in the formation of CNV, thereby avoiding systemic adverse effects.

Lower drug resistance: Anti-VEGF therapy is prone to drug resistance because other growth factors can compensate for the effects of VEGF. Targeting metabolic pathways makes it difficult for changes in growth factor levels to affect therapeutic efficacy.

Large potential for combination therapy: EC metabolic regulation can be combined with anti-VEGF and other treatments, and exploration of targeting multiple metabolic pathways simultaneously may enhance therapeutic effects. Some combination therapy strategies have demonstrated good synergistic effects in animal models, such as 3PO in combination with anti-VEGFR2 monoclonal antibody DC101 to enhance the anti-CNV effect in mice [18]; terpestacin in combination with bevacizumab showed stronger synergistic anti-tumor angiogenesis [84]. However, research in this area is still in its infancy and more exploration is needed in the future.

## 5. Conclusions and Perspectives

Traditionally, the treatment of wet AMD has primarily focused on anti-VEGF therapy. Over the past 20 years, anti-VEGF treatment has made major progress but has also revealed some issues such as treatment ineffectiveness or resistance in some patients, prompting researchers to seek new therapeutic strategies. Meanwhile, with in-depth research in recent years, the importance of EC metabolic regulation as a new therapeutic strategy for wet AMD has become increasingly clear. Specifically, precise regulation of specific metabolic pathways in ECs can change their phenotype and angiogenic function through metabolic reprogramming, independent of exogenous growth signals. This suggests that EC metabolism itself can impact its participation in neovascularization, challenging the conventional view that metabolism is merely a passive response to growth stimulation. Some scholars have likened angiogenesis to driving a car, with growth factors as the driver and EC metabolism as the engine [12]. Blocking the driver allows replacing the driver, but blocking the engine will prevent the car from moving, clearly illustrating the key role of EC metabolism in angiogenesis. However, current related research is still at an early stage, with few clinical studies. The underlying mechanisms require further elucidation, and clinical application pathways need to be further clarified. Future research should explore in greater depth the role of EC metabolism in the pathogenic mechanisms of wet AMD and conduct more preclinical studies and preliminary clinical trials to evaluate safety and effectiveness. It is also worth exploring the potential for combined targeting of multiple metabolic pathways or use in combination with anti-VEGF therapy to optimize treatment outcomes. Metabolic regulation of ECs is likely to emerge as a novel important therapeutic strategy for wet AMD as research continues to advance in depth.

## Figures and Tables

**Figure 1 ijms-25-05926-f001:**
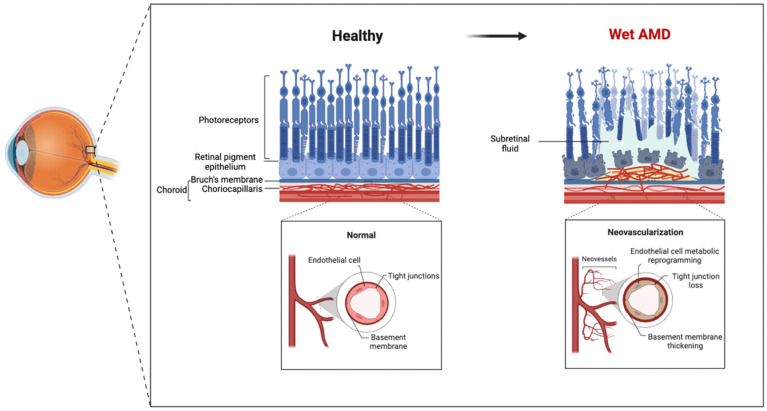
Schematic illustration of the retinal structure in healthy state and wet AMD state. In the healthy retina, the retinal pigment epithelium (RPE) layer and the choroid are tightly connected, forming the intact Bruch’s membrane, which helps maintain the structural and functional integrity of the retina. The retinal neurons, including photoreceptor cells (cone and rod cells), are arranged in an orderly manner, and the integrity of the retinal vasculature is maintained. In contrast, in the retina of wet AMD patients, the tight connection between the RPE and the choroid is disrupted. CNV breaches Bruch’s membrane and grows towards the retina, leading to retinal edema and structural disorganization. Additionally, the leakage from CNV further exacerbates retinal edema, and the structure and function of retinal neurons, particularly the cone cells, are consequently impaired. These pathological changes ultimately result in severe central vision loss in wet AMD patients. ECs serve as the effector cells in the processes of CNV formation and vascular leakage, and their metabolic dysregulation plays a crucial role in these pathological events.

**Figure 2 ijms-25-05926-f002:**
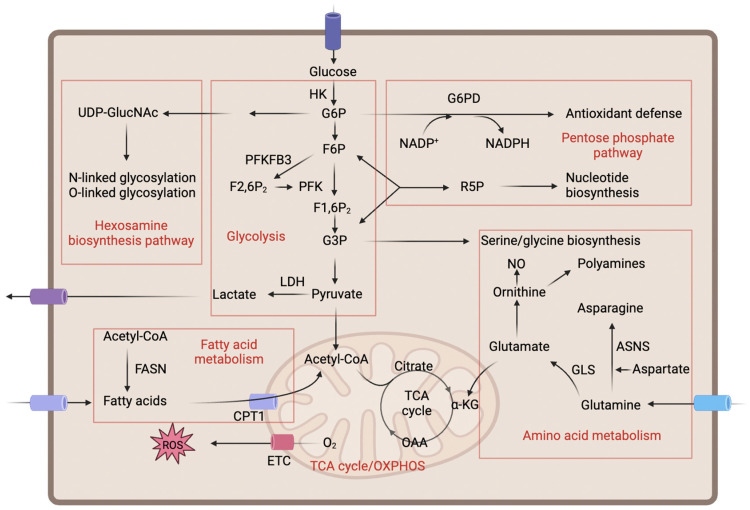
General metabolism in ECs. Individual metabolic pathways and names are highlighted with red boxes and characters. The arrows in the figure represent the interconversions between metabolites and the direction of metabolic flux. Abbreviations used: α-KG: α-ketoglutarate; ASNS: asparagine synthetase; CPT1: carnitine palmitoyltransferase 1; ETC: electron transport chain; F1,6P_2_: fructose-1,6-bisphosphate; F2,6P_2_: fructose-2,6-bisphosphate; F6P: fructose-6-phosphate; FASN: fatty acid synthase; G3P: glyceraldehyde-3-phosphate; G6P: glucose-6-phosphate; G6PD: glucose 6-phosphate dehydrogenase; GLS: glutaminase; HK: hexokinase; LDH: lactate dehydrogenase; NADP: oxidized nicotinamide adenine dinucleotide phosphate; NADPH: reduced nicotinamide adenine dinucleotide phosphate; NO: nitric oxide; OAA: oxaloacetate; OXPHOS: oxidative phosphorylation; PFK: phosphofructokinase; PFKFB3: 6-phosphofructo-2-kinase/fructose-2,6-bisphosphatase 3; R5P: ribose-5-phosphate; ROS: reactive oxygen species; TCA: tricarboxylic acid; UDP-GlucNAc: uridine diphosphate N-acetylglucosamine.

**Table 1 ijms-25-05926-t001:** Targeting EC metabolism for therapeutic strategies.

Metabolic Pathways	Targets	Compounds	Effects	References
Glycolysis	PFKFB3	3PO	Inhibition of pathological angiogenesis in CNV and ROP models	[18]
		PFK-15	Inhibition of infantile hemangioma angiogenesis	[77]
		PFK-158	Anti-cancer and anti-atherosclerosis effects	[78,79]
	HK2	2-DG	Reduced HUVEC angiogenesis and anti-tumor angiogenesis	[80,81]
	PKM2	Shikonin	Instability of EC junctions and weakened angiogenesis	[82]
Mitochondrial Metabolism	Mitochondrial respiratory chain	Terpestacin	Inhibition of angiogenesis in zebrafish and anti-tumor angiogenesis	[83,84,85]
	oxidative stress	MitoQ	Improved vascular lesions in OIR and STZ mouse models	[86]
Lipid metabolism	FASN	Orlistat	Reduced vascular tuft formation in ROP model and anti-tumor angiogenesis	[87,88]
	CPT1	Etomoxir	Inhibition of retinal EC proliferation and pathological angiogenesis in ROP model	[43]
	HMG-CoA	Pitavastatin	Anti-angiogenic effects on CNV in rats and mice	[89,90]
Pentose phosphate pathway	G6PD	6-AN	Inhibition of angiogenic response and anti-tumor effects	[91,92]
Glycosylation	VEGFR2	Itraconazole	Anti-angiogenic effects on various ECs of different origins in vitro, and inhibition of CNV development in rats	[93,94,95]
Glutaminolysis	GLS	DON, BPTES, CB839	Inhibition of angiogenesis in various EC sources and reduced vascular tuft formation in ROP model	[21,96]

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
