# Peer review of "Metabolic Regulation of Endothelial Cells: A New Era for Treating Wet Age-Related Macular Degeneration"

_ijms, 2024, doi:10.3390/ijms25115926_

Round 1

Reviewer 1 Report

Comments and Suggestions for Authors

Age-related macular degeneration (AMD), a degenerative condition of the retina,

emerges as the predominant cause of visual impairment and profound vision decline,

particularly in the aged demographic[1, 2]. Forecasts indicate a surge in AMD prevalence, with patient numbers anticipated to escalate from nearly 200 million in

2020 to around 300 million by 2040[3]. AMD manifests in two primary forms: dry and

wet, with the latter, though less prevalent, being chiefly responsible for the significant

instances of visual deterioration. The pathogenesis of wet AMD is intricate, currently

understood to be the result of genetic, age-related, and environmental interplay.

The topic presented by the authors is very interesting and very worrying for new therapies for wet AMD.

After reading the complete manuscript, I have observed that the literature consulted for this manuscript is good with 125 citations, although the authors could have further enriched the manuscript with new, more up-to-date citations on AMD.

In relation to these observations, I have a few questions:

1. Can the authors explain the biochemical mechanism of the enzyme 6-phosphofructo-2-kinase/fructose-2,6-bisphosphatase 3 (PFKFB3) in AMD on regulation of angiogenesis?

2. How do endothelial cells participate in new therapies for wet AMD?

3. Could you explain the pharmacological mechanism of choroidal neovascularization (CNV) for the treatment of wet AMD?

4. Orlistat is a selective FASN inhibitor that is mainly used to treat obesity. What relationship can explain the authors' use of Orlistat for the treatment of wet AMD?

5. What relationship exists between lipid metabolism and future treatments for the retina and very especially for wet AMD.

6. Traditionally, the treatment of wet AMD has focused primarily on anti-VEGF therapy. Over In the last 20 years, anti-VEGF treatment has made great progress but has also revealed some problems, as the authors raise, now my question is what do the authors recommend to solve the problems of adverse reactions of anti-VEGF therapy to search for new therapeutic strategies for AMD?

Reviewer 2 Report

Comments and Suggestions for Authors

Summary:

Interest in the role of endothelial cell metabolism as a regulator of angiogenesis is gaining marked increases in attention.  The review describes studies showing that it is likely that continued effort is likely to generate novel therapy for improved management of wet AMD in a clinical setting.  This is a promising approach because most metabolic enzymes are druggable, and endothelial cell metabolism increasingly appears to differ from the metabolism in other cell types. The well written review summarizes different approaches that resolve the active roles of endothelial metabolic regulation in angiogenesis. In disease states such as wet AMD, disturbances of metabolic pathways may lead to excessive neovascularization.  The authors present results showing that targeted modulation of endothelial metabolism is a promising therapeutic strategy for wet AMD.   The potential applications of endothelial metabolic regulation in wet AMD treatment are dealt with from multiple perspectives.  The major endothelial metabolic pathways, and novel therapeutic approaches are described in depth that target altered metabolism in wet AMD. Even though the literature citations are adequate, it is worthwhile to also cite PMID 32833569. (Basic and Therapeutic Aspects of Angiogenesis Updated)

Author Response

Thank you for the positive feedback and insightful suggestion. Upon careful reading, we found the recommended article (PMID 32833569) to be highly relevant to our content. We have incorporated this reference into the revised manuscript. (page 2, line 62-64)

Reviewer 3 Report

Comments and Suggestions for Authors

In the manuscript entitled  ¨Metabolic Regulation of Endothelial Cells: A New Era for Treating Wet Age related Macular Degeneration¨, author describes in an extensive form the potential application of Endothelial Cells in metabolic regulation in wet AMD. Also, the authors discuss the EC intracellular pathways to provide a novel targets for a therapeutic approach. The manuscript is well writing.

The manuscript could be accepted in the present form.

Author Response

Thank you for the positive feedback and recommendation to accept the manuscript in its current form. We appreciate the reviewer's time and expertise in evaluating this work.

Reviewer 4 Report

Comments and Suggestions for Authors

The manuscript titled “Metabolic Regulation of Endothelial Cells: A New Era for Treating Wet Age related Macular Degeneration”, is a well thought out review article. Basing on the current literature, the authors tried to link metabolic disturbances of Endothelial cells (ECs) to angiogenesis and excessive neovascularization that is seen in retinal degenerative diseases such as wet AMD. Manuscript writing is excellent and satisfactory. Further I haven’t found any deviations in language or grammatical errors. My only suggestion for authors is to try to expand in detail some important abbreviations (such as YAP, FASN etc)…either authors could draft a new abbreviation table or expand the names of the abbreviations in the text itself. I also suggest the authors to improve the text in the figures…text (font size) in the figures is barely visible…Leaving these two points, the review article is written excellently and I have found no unnecessary references or self-citations and I suggest the Editors to accept the manuscript in the present form with minor modifications.  

Author Response

We appreciate the reviewer's positive feedback and constructive suggestions. As recommended, we have increased the font size in Figure 1 by 1-3 points to enhance readability. As for the abbreviations, we have carefully reviewed the manuscript and expanded the names of abbreviations that were not defined upon first mention, such as YAP. These changes have been highlighted in the revised manuscript.